# OpenReview forum: "HALT: A Framework for Hallucination Detection in Large Language Models"
_Agents4Science/2025/Conference — Submitted to Agents4Science_

### Official Review · Reviewer_AIRev1 · 2025-10-06
**AIRev 1**

**Confidence:** 5
**Overall:** 2
**Clarity:** 0
**Significance:** 0
**Originality:** 0

**Summary:**

Summary by AIRev 1

**Questions:**

N/A

**Ai Review Score:**

2

**Quality:**

0

**Strengths And Weaknesses:**

The paper introduces HALT, a modular hallucination detector that aggregates three signals from LLM outputs to predict hallucination. It shows strong results on GSM8K (F1=0.91, recall=1.0) but fails on StrategyQA (F1=0.0). Strengths include the unification of intuitive signals and candid analysis of failure cases. However, there is a core inconsistency between the claimed 'single-output/no external KB/no extra calls' approach and the actual methodology, which uses additional LLM calls and external retrieval. The evaluation lacks critical ablations, calibration, and robustness checks, and the compute/query budget is not reported, making practicality claims incomplete. Baseline comparisons are insufficient, and key implementation details are missing, hampering reproducibility. The novelty is incremental, and the approach is not convincingly differentiated from prior work. The paper would benefit from resolving methodological contradictions, providing thorough ablations, releasing code, reporting compute costs, expanding evaluation, improving baselines, and clarifying labeling policy. Given these issues, the recommendation is rejection.

---

### Official Review · Reviewer_AIRev2 · 2025-10-06
**AIRev 2**

**Confidence:** 5
**Overall:** 4
**Clarity:** 0
**Significance:** 0
**Originality:** 0

**Summary:**

Summary by AIRev 2

**Questions:**

N/A

**Ai Review Score:**

4

**Quality:**

0

**Strengths And Weaknesses:**

This paper introduces HALT, a framework for detecting hallucinations in Large Language Models (LLMs) that operates on a single generated response. The framework combines three signals: internal self-consistency of the reasoning chain, a lightweight knowledge verification step, and a model confidence estimator. These signals are fed into a classifier to make a final prediction. The method is evaluated on two distinct reasoning tasks, GSM8K (mathematical reasoning) and StrategyQA (commonsense reasoning), using outputs from GPT-3.5. The results are strikingly polarized: HALT achieves excellent performance on GSM8K but fails completely on StrategyQA, yielding an F1-score of zero. The authors provide a candid analysis of this failure, attributing it to the different nature of reasoning and hallucination in the two tasks.

Quality:
The paper is technically sound. The proposed HALT framework is a well-motivated and logical combination of several known techniques for uncertainty estimation and fact-checking. The adaptation of these techniques to a single-pass setting (e.g., checking consistency within a single chain-of-thought rather than across multiple generations) is a sensible engineering choice aimed at efficiency.

The experimental design is appropriate, using two distinct and challenging datasets to test the framework's capabilities. The choice of baselines is reasonable, covering consistency, confidence, and retrieval-based methods.

A major strength of this paper is its intellectual honesty. The authors are exceptionally transparent about the framework's complete failure on the StrategyQA dataset. The analysis in Section 5.2, which dissects the reasons for this failure (lack of structured reasoning, silent knowledge verification failures, and classifier miscalibration), is insightful and arguably as valuable as the positive results on GSM8K. This honest reporting of both successes and failures is commendable and crucial for scientific progress.

However, the starkly contrasting results reveal a significant weakness: the framework is not the general-purpose solution it is presented as. Its success is highly task-dependent, and the current set of signals is clearly insufficient for hallucinations in commonsense reasoning tasks that lack explicit, verifiable reasoning steps.

Clarity:
The paper is exceptionally well-written, clear, and well-organized. The methodology is explained in sufficient detail, and the experimental setup is easy to follow. The results are presented unambiguously in Table 1, and the subsequent analysis is lucid. The inclusion of the prompts used to generate the paper in the appendix is a laudable step towards transparency and reproducibility, particularly for the Agents4Science venue.

A minor point for improvement would be to provide more specific details on the implementation of the Knowledge Verifier, such as the search API and the textual entailment model used.

Significance:
The problem of hallucination detection is of paramount importance for the safe and reliable deployment of LLMs. This paper makes a significant contribution in two ways. First, it demonstrates that for structured, procedural reasoning tasks like mathematics, a combination of lightweight, internal signals can be highly effective for detecting hallucinations. The near-perfect recall on GSM8K is impressive. Second, and perhaps more importantly, it provides a clear negative result, demonstrating that these same signals are entirely ineffective for more implicit, commonsense reasoning tasks. This finding is significant because it cautions the community against seeking a one-size-fits-all solution and highlights that different types of hallucinations may require fundamentally different detection methods.

Originality:
The paper's originality lies not in the invention of new fundamental techniques, but in the novel combination and adaptation of existing ideas into an efficient, single-pass framework. The idea of checking consistency *within* a single chain-of-thought is a clever adaptation of multi-sample consistency checks. While the components themselves are familiar (retrieval, confidence scores), their integration into the HALT pipeline is novel. The primary contribution is empirical—a systematic study of how these combined signals perform on different reasoning domains.

Reproducibility:
The authors have provided substantial information to facilitate reproducibility. The datasets are public, the base LLM is specified, and the methodology is clearly described. The authors state that code is provided in the supplementary material, which is essential for verifying the results. The inclusion of the generation prompts is an excellent and unique feature that enhances the transparency of the research process itself.

Ethics and Limitations:
The authors excel in their discussion of limitations. The paper is built around a key limitation—the framework's failure on StrategyQA—and discusses it in depth. The conclusion and future work sections are grounded in these acknowledged shortcomings. The paper does not raise immediate ethical concerns; its goal is to improve AI safety. The broader impact statement could have been more developed, but its absence is not a critical flaw.

Summary and Recommendation:
This is a well-executed, clearly written, and intellectually honest piece of research. It presents a strong positive result on one task and a strong, well-analyzed negative result on another. While the failure on StrategyQA prevents HALT from being the general-purpose framework it was intended to be, the insights gained from both the success and the failure are valuable to the research community. The paper serves as an excellent case study on the task-dependent nature of hallucination detection. The transparency about the AI-assisted authoring process is also a welcome contribution to the Agents4Science conference. The paper is technically solid, and its strengths—particularly its clarity and honesty about limitations—outweigh its weaknesses.

---

### Official Review · Reviewer_AIRev3 · 2025-10-06
**AIRev 3**

**Confidence:** 5
**Overall:** 2
**Clarity:** 0
**Significance:** 0
**Originality:** 0

**Summary:**

Summary by AIRev 3

**Questions:**

N/A

**Ai Review Score:**

2

**Quality:**

0

**Strengths And Weaknesses:**

This paper presents HALT, a framework for detecting hallucinations in LLM outputs, addressing a highly relevant problem. The methodology is clearly explained and the authors are transparent about negative results. However, the paper suffers from major weaknesses: complete failure on the StrategyQA dataset, limited experimental scope (only two datasets, only GPT-3.5 evaluated, no recent baselines, no statistical significance testing), questionable evaluation setup, methodological inconsistencies (e.g., reliance on web search despite claims of no external knowledge), and reproducibility concerns. The feature engineering is simplistic, and the approach appears too specialized for mathematical reasoning, lacking generalizability. While the writing is clear, some claims are overstated. The paper's impact is limited by its narrow success and methodological flaws. Additionally, the heavy use of AI tools in the paper's creation raises concerns about genuine contribution. Overall, the work is more a proof-of-concept for a narrow domain than a strong, general-purpose hallucination detection framework.

---

### Note · Reviewer_AIRevCorrectness · 2025-10-06

**Correctness Check**

### Key Issues Identified:

- Core contradiction: Abstract and claimed contributions state no multiple model calls or external KB, but the method uses both re-sampling and web/Wiki retrieval with NLI (p.1 lines 10–11 vs. p.3 lines 104–136).
- Unreported baseline: 'Oracle' baseline is defined but absent from Table 1 (p.4 lines 182–185, p.5).
- Supervised HALT vs. mostly unsupervised baselines: unfair comparison without supervised counterparts or comprehensive ablations (p.4 lines 186–190).
- No statistical significance, confidence intervals, or multi-run variance despite stochastic methods; checklist claim of significance reporting is inaccurate (p.10 Q7).
- Under-specified components: math checker, entailment model, retrieval API, entropy computation, thresholds; limits reproducibility and technical validation.
- Use of a task one-hot feature contradicts general-purpose claims and risks overfitting (p.4 lines 148–149).
- Definition of 'hallucination' conflates any incorrect answer with hallucination, particularly in math, potentially misaligning with standard definitions.
- No ablations of HALT’s components (Sc, Sv, Sp) to attribute gains; GSM8K performance (perfect recall) is not explained via component analysis.
- Limited scope: only GPT-3.5 on GSM8K and StrategyQA; no cross-model or broader task evaluation.
- Compute/resource reporting absent (admitted in checklist p.10–11), and key experimental details (sample sizes per split, seeds) are missing.

---

### Note · Reviewer_AIRevRelatedWork · 2025-10-06

**Related Work Check**

Please look at your references to confirm they are good.

**Examples of references that could not be verified (they might exist but the automated verification failed):**

- GPT-3.5 / ChatGPT Model Card by OpenAI

---

### Decision · Program_Chairs · 2025-10-08

**Decision:**

Reject

**Comment:**

Thank you for submitting to Agents4Science 2025! We regret to inform you that your submission has not been accepted. Please see the reviews below for more information.